

# MATRIX-VBS: implementing an evolving organic aerosol volatility in an aerosol microphysics model

Chloe Y. Gao[1,2], Kostas Tsigaridis[3,2], Susanne E. Bauer[3,2]

[1]Department of Earth and Environmental Sciences, Columbia University, New York, NY, 10025, USA
[2]NASA Goddard Institute for Space Studies, New York, NY, 10025, USA
[3]Center for Climate System Research, Columbia University, New York, NY, 10025, USA

*Correspondence to*: K. Tsigaridis (kostas.tsigaridis@columbia.edu)

**Abstract.** We have implemented an existing aerosol microphysics scheme into a box model framework and

extended it to represent gas-particle partitioning and chemical ageing of semi-volatile organic aerosols. We then applied this new research tool to investigate the effects of semi-volatile organic species on the growth, composition and mixing state of aerosol particles in case studies representing several different environments. The volatility-basis set (VBS) framework is implemented into the aerosol microphysical scheme MATRIX (Multiconfiguration Aerosol TRacker of mIXing state), which resolves mass and number aerosol

concentrations and in multiple mixing-state classes. The new scheme, MATRIX-VBS, has the potential to significantly advance the representation of organic aerosols in Earth system models by improving upon the conventional representation as non-volatile particulate organic matter, often with also an assumed fixed size distribution. We present results from idealized cases representing Beijing, Mexico City, a Finnish and a Southeast U.S. forest, and investigate the evolution of mass concentrations and volatility distributions for

organic species across the gas and particle phases, as well as assessing their mixing state among aerosol populations. Emitted semi-volatile primary organic aerosols evaporate almost completely in the high volatile range, while they remain in the particle phase in the low volatility range. Their volatility distribution depends on the applied emission factors, oxidation by OH radicals, and temperature. We also compare against parallel simulations with the original scheme, which represented only the particulate and non-volatile component of

the organic aerosol, examining how differently the condensed phase organic matter is distributed across the mixing states in the model. The results demonstrate the importance of representing organic aerosol as a semi-volatile aerosol, and explicitly calculating the partitioning of organic species between the gas and particulate phases.

Keywords: organic aerosols, volatility-basis set, aerosol mixing state, box model

## 1 Introduction

Atmospheric aerosols play a key role in the Earth system with great impacts on global air quality, public



health and climate (Boucher et al., 2013; Myhre et al., 2013; Seinfeld and Pandis, 2016). One contribution to the large uncertainty in aerosol radiative forcing is organic aerosol (OA), which is ubiquitous in the atmosphere and contribute to a large portion of submicron particulate mass in various regions around the world (Zhang et al., 2007; Jimenez et al., 2009). Advancements in measurement techniques greatly improved

our understanding of the evolution of OA and lifetime in the atmosphere at the process level (Jimenez et al., 2009). However, OA processes in models still remain poorly constrained. Measurements imply that OA concentrations are potentially underestimated in current models (Tsigaridis et al., 2014). Such a discrepancy hints at large uncertainties in the prediction of aerosol-radiation interactions, their hygroscopicity, aerosol-cloud interactions and their overall impact on climate (Petters and Kreidenweis, 2007).

Missing sources of secondary organic aerosol (SOA) in models have been suggested to be the main cause of the underestimated OA formation (Heald et al., 2005; Volkamer et al., 2006; Hodzic et al., 2010; Spracklen et al., 2011). More recently, studies have sought to investigate the underestimation of organic aerosol mass within more advanced model frameworks, which are capable of resolving  semi-volatile primary organic aerosol (POA) and including secondary organic aerosol (SOA) from a wider set of precursors including

intermediate volatility organic compounds (IVOCs). The volatility-basis set was developed (Donahue et al., 2006) to provide a relatively simple framework whereby models can represent the overall behavior of the myriad of compounds that constitute organic aerosol and their precursors. The approach involves considering OA as being composed of a number of representative species, each with a particular volatility, spanning a spectrum in vapor pressures from highly volatile (which essentially remains in the gas phase) to very low

vapor pressure species which partition readily into the particle phase. VBS then captures the chemical ageing of the organic species in the gas phase, with hydroxyl radical oxidizing them and producing the adjacent lower volatility class as a product. This method has been used extensively in regional studies (Robinson et al., 2007; Shrivastava et al., 2008; Murphy and Pandis, 2009; Tsimpidi et al., 2010; Hodzic et al., 2010; Fountoukis et al., 2011; Tsimpidi et al., 2011; Bergström et al., 2012; Athanasopoulou et al., 2013; Zhang et al., 2013;

Fountoukis et al., 2014) but less so in global models (Pye and Seinfeld, 2010; Jathar et al., 2011; Jo et al., 2013; Tsimpidi et al., 2014; Hodzic et al., 2015). Other studies have used the 2D-VBS (Donahue et al., 2011; Murphy et al., 2011), an approach that on top of the volatility space it also resolves that of chemical composition, by tracking the amount of oxygenation in the representative organic compounds. However, it is not implemented in global models, due to its large amount of tracers and the large number of free parameters

that are involved in the parameterization.

The inclusion of semi-volatile organics is important for accounting for the total mass of organics in the particulate phase, since an increase in particulate organic matter may not be the result of chemically produced low volatility species, but simply be reflecting a temperature-driven increase in the partitioning of semi-

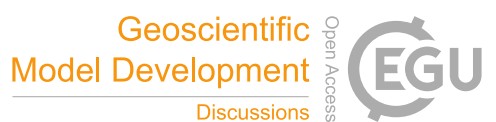

volatile organic aerosol into the particle phase. It has been established that the very low volatility organics play a key role in particle growth, while the range of volatilities contributing to aerosol growth increases with aerosol size (Pierce et al., 2011; Yu, 2011). Semi-volatile organics also affect aerosol size and mixing state, as well as their impact on climate, due to changes in cloud condensation nuclei (CCN) formation rates (Petters et

al., 2006, Riipenen et al., 2011; Scott et al., 2015), hygroscopicity, and new particle formation (Metzger et al., 2011; Paasonen et al., 2013). Since OA emissions are on the rise from developing countries (Lamarque et al., 2010) and no Earth system model considers anthropogenic OA as semi-volatile as measurements suggest, it is important to include and constrain semi-volatile organics to ultimately reduce uncertainties in aerosol radiative forcing and make climate model simulated aerosol changes more realistic.

The objective of this study is to further develop an aerosol microphysics model by including a more advanced representation of organic aerosol, including semi-volatile primary OA and an evolving OA volatility during chemical ageing in the gas phase in its calculations. This objective was achieved by implementing the VBS framework in the aerosol microphysical scheme MATRIX (Bauer et al., 2008), which represents major aerosol processes such as nucleation, condensation (excluding organics in its original version) and coagulation, and

explicitly tracks the mixing state of different aerosol populations. As many traditional chemistry-climate models do (Tsigaridis et al., 2014), MATRIX treats POA and SOA as non-volatile (Bauer et al., 2008). By coupling MATRIX with VBS, POA are treated as condensable semi-volatile organic compounds. These can partition among different aerosol populations based on their volatility and aerosol population size distribution, capturing particle growth via condensation of low volatility organic vapors, thus providing a more physically-

based calculation of aerosol microphysics.

The semi-volatile nature of biogenic SOA is not represented in the VBS framework in this work. Instead, biogenic SOA are treated as non-volatile, as in the original MATRIX version, and are produced with a 10% constant yield from terpenes emissions without any requirement for oxidation before the OA is formed (Lathière et al., 2005; Tsigaridis et al., 2014). The inclusion of semi-volatile biogenic SOA will be

parameterized in the same way as in the VBS framework presented here in the future.

## 2 Model description

A box model is used for this study. The gas phase chemical mechanism CBM-IV (Gery et al., 1989), as used in the NASA GISS ModelE (Shindell et al., 2001; Shindell et al., 2003), is coupled to the MATRIX aerosol microphysics scheme, utilizing the Kinetic Pre-Processor KPP (Sandu and Sander, 2006) to solve the

differential equations of the gas phase chemistry scheme.



### 2.1 MATRIX box model

MATRIX (Multiconfiguration Aerosol TRacker of mIXing state; Bauer et al., 2008) is an aerosol microphysical model based on the Quadrature Method of Moments scheme (McGraw, 1997) in the NASA GISS ModelE Earth System Model, which can be used either as a module within the global model or as a

stand-alone box-model. Here, the stand-alone box model is used for development. The design of the code is such that the box-model code can be used as-is in the global model, without any changes, allowing for seamless transition and maximum portability. MATRIX is designed to resolve the aerosol temporal evolution and represent the mixing states of a user-selected set of aerosol populations. It describes new particle formation, particle growth through condensation with explicit treatment of sulfuric acid condensation and

lumped treatment of the $NH_4$-$NO_3$-$H_2O$ system, as well as coagulation of particles among different populations. Each aerosol population has its own set of aerosol components, which may be primary (from direct aerosol emissions), secondary (formed by nucleation or condensation of gas phase components onto existing primary particles), or mixed (from any constituent, following condensation on primary aerosols or coagulation between primary/secondary/mixed populations).

### 2.2 VBS framework

The volatility-basis set approach is introduced to the original model; it is an organic aerosol volatility parameterization that separates semi-volatile organic compounds into logarithmically-spaced bins of effective saturation concentrations, which are used for gas-particle partitioning and photochemical aging (Donahue et al., 2006). The scheme groups organic compounds into nine surrogate VBS species according to their effective

saturation concentrations ($C^*$) at 298 K, which are separated by factors of ten, ranging from $10^{-2}$ to $10^6$ µg m$^{-3}$. Species in the $10^{-2}$ µg m$^{-3}$ bin are the least volatile ones and partition almost exclusively to the particulate phase, while species in the $10^6$ µg m$^{-3}$ bin are the most volatile ones and remain almost exclusively in the gas phase. These compounds can become chemically aged while in the gas phase following reaction with •OH radicals, which results in oxidizing to a species with a factor of 10 lower volatility (Donahue et al., 2006).

Parameters and names used to represent them in this study are listed in Table 1.

### 3 Model development

In the original version of the MATRIX model, organics only contribute to particle growth and mix with other aerosol species via coagulation. Primary organic aerosols are emitted only as non-volatile particulate organic matter, and do not exist in the gas phase or interact with other aerosol populations. Implementing the VBS

scheme adds these missing processes. Before this development, there were eight alternative configurations of MATRIX available to the user, each representing a distinct set of aerosol populations whose number,



composition and interactions by coagulation vary. A ninth configuration with 15 selected aerosol populations is created for this study (Table 2), which builds on top of configuration 1 in Bauer et al., 2008, in which eight of the 14 populations, ACC, OCC, BC1, BC2, OCS, BOC, BCS, and MXX, are set to contain organics as semi-volatile VBS species. We only included semi-volatile organics in eight populations, so that we can

examine the BC-OA-sulfate-nitrate system first, before adding them into the nucleation population AKK and the dust and sea salt populations (DD1, DS1, DD2, DS2, SSA, SSC). Through coagulation, the fifteen donor populations grow or mix and are placed into recipient populations, based on the donor population composition, as described in Bauer et al. (2008). In a future stage, organics will also be implemented in the AKK mode to present nanoparticle growth and we will include an additional nucleation scheme that considers the

dependence of new particle formation that involve organics (Kirkby et al. 2016, Tröstl et al. 2016).

Previously, each aerosol population carried up to 5 tracers – sulfate, black carbon, nonvolatile organics, dust and sea salt. Now the eight organic-containing populations carry 9 additional semi-volatile VBS species listed in Table 1, to up to 14 available tracers per population, with the original organics tracer representing non-volatile OA, as it did in the original mechanism. This newly coupled model MATRIX-VBS treats POA as

semi-volatile gas phase species, which then partition into and out of the particulate phase.

## 4 Simulations

To test the newly developed model's behavior, we simulated idealized cases representative of four different locations of different environments: one very polluted city (Beijing), another cleaner yet still very polluted city at high altitude (Mexico City) and closer to the tropics, a very clean Finnish forest (Hyytiälä), and an

anthropogenically-affected forest in the Southeast U.S. (Centreville, Alabama). The experiments are performed for a winter (January) and a summer month (July) for ten days, and initial conditions and emission rates for each location were extracted from a GISS ModelE simulation (similar setup as described in Mezuman et al., 2016) for the year 2006, listed in Table 3, with no deposition and dilution, for simpler mass-balance calculations. Semi-volatile POA, sulfate in the accumulation mode, and black carbon, are emitted in

the OCC, ACC, BC1 populations, respectively, shown in Figure 1 as yellow circles. The emission rates for the VBS species were derived from the POA emission rate in the global model for the corresponding gridbox and month, which were distributed in the volatility space by using mass-based emission factors from Shrivastava et al. 2008 (Table 1). Adding up the 9 factors from each bin listed in Table 1, we obtain a total factor of 2.5, which means the new scheme's organics emission is 2.5 times that of the organics emissions in the original

scheme. The additional multiplication factor of 1.5 is applied to the emission to account for missing sources of volatile organics in the IVOC volatility regime in the inventories. Hodzic et al. (2015) showed that

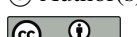

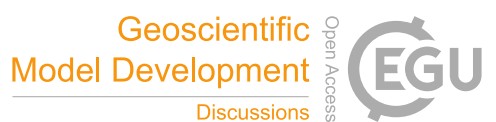

Shrivastava et al. (2008) overestimates biomass burning, which could affect the emission factors we used, however, it is not an issue for the idealized experiments in the present study.

Black carbon is uniquely treated in MATRIX, in order to separate the coated (via condensation) from the mixed (via coagulation) populations. It is emitted in BC1, which can grow (blue arrow in Figure 1) with

inorganic and organic coating, and as its coating volume fraction reaches 5%, it would be moved in the BC2 population, shown as the orange circle. The four organic-containing populations described above can coagulate (black arrows in Figure 1) with themselves and each other and form three additional organic-containing mixed populations, BOC, OCS and BCS, shown as green circles. This schematic includes seven of the eight organic-containing populations in the model.

**5 Results and discussion**

The temporal evolution of the total organics mass concentrations from the new scheme and the old scheme are presented in Figure 2 for January and Figure 3 for July in the four locations under study. They show large changes in organics concentrations between the old scheme (black line on the right column) and the new one (colors). The organics in the new scheme are represented and distributed by organic tracers of different

volatility, whose saturation concentration C* ranges from the least volatile $10^{-2}$ μg m$^{-3}$ ("M2" in Figures 2 and 3) to the most volatile $10^{6}$ μg m$^{-3}$ ("P6" in Figures 2 and 3). They are distributed between the gas and aerosol phases by gas-particle partitioning, whereas the organics in the original scheme are only represented by one nonvolatile organic aerosol tracer ('OCAR').

As mentioned in the model description, the emission rates for organics in the each of the volatility bin in the

new scheme were derived from the mass-based emission factors, of 2.5, from Shrivastava et al. 2008. Consequently, since there is no deposition and dilution in the simulations, the new scheme's organics total mass concentrations (shown in color in the right columns of Figures 2 and 3) always adds up to 2.5 times that of the old scheme (shown as dash-dotted lines) throughout the simulations in both January and July.

**5.1 Winter**

In January, the total mass concentration for organics in Beijing, Centreville, Hyytiälä and Mexico City at the end of 10 days are about 115 μg/m$^{3}$, 16 μg/m$^{3}$, 13 μg/m$^{3}$, and 65 μg/m$^{3}$, respectively. Organic VBS species partition between the gas and aerosol phases within their corresponding volatility bin. The more volatile the species, the more it partitions into the gas phase. The concentration evolution of VBS species in the gas phase from the four locations are shown in the left column of Figure 2 for January. From top to bottom in each panel,

volatility decreases from the most volatile species ("P6") to the least one ("M2"). Although semi-volatile organics are emitted in the aerosol phase, in the high volatility range from P6 to P4 bins, the species are so





volatile that they evaporate and partition into the gas phase almost completely. Volatile organics are oxidized in the gas phase by the extremely reactive hydroxyl radicals (˙OH) during daytime, and as they become more oxidized, their volatility decreases and they move down to the adjacent lower volatility bin. This oxidation process is called "aging" in the VBS framework. For example, species from the P6 bin are emitted in the

aerosol phase, but because they are so volatile, they evaporate immediately into the gas phase. During daytime, when OH radicals are abundant, they age, become less volatile, and are placed in the lower volatility bin P5, which has a C* one order of magnitude lower than its precursor. Similarly, as gas phase species in all volatility bins age, they get oxidized, become less volatile, and move further down the volatility space. At intermediate volatilities species are becoming less volatile, and they no longer completely evaporate after

emission, but are present in both the gas and the aerosol phases, following partitioning. However, they continue to become even less oxidized following oxidation by OH, until they have such low volatility that they almost exclusively partition into the aerosol phase (Figure 2 middle column).

In all four locations, almost all species in the high volatility range are in the gas phase, those in the intermediate volatility range partition between the gas and aerosol phases, and those in the low volatility range

are in the aerosol phase in January. This is especially true for Beijing and Hyytiälä, where the volatility distributions are very similar (in relative terms), where the total concentration of gas phase species is higher than the sum of all aerosol-phase species. In Centreville, the total amount of gas phase species is about the same with that of the aerosol phase species, whereas in Mexico City there are more species in the aerosol than in the gas phase. In Centreville and Mexico City the species show a diurnal variability, which will be

explained later.

Aging can help explain the similar volatility distributions in Beijing and Hyytiälä. The ˙OH concentration in both locations are low in January: Beijing's mean ˙OH is about $10^5$ molecules/cm$^3$ and Hyytiälä's mean ˙OH is about $10^4$ molecules/cm$^3$. Low ˙OH concentrations limit the aging of high volatility organics and their ability to move to the lower volatility bins, thus the volatility distributions do not change drastically, something that

is also evident by the lack of a daily cycle. On the other hand, much higher mean ˙OH concentrations in Centreville ($2*10^6$ molecules/cm$^3$) and Mexico City ($5*10^6$ molecules/cm$^3$) provide more oxidation power, making oxidation a significant pathway in aerosol evolution. The higher mean ˙OH concentrations also explain the diurnal variability of both gas phase and aerosol phase mass concentrations we see in the two locations, because ˙OH is only produced during daytime and has very low concentrations during night. Since Mexico

City has slightly higher ˙OH concentration than Centreville, its total gas phase reaches a dynamic equilibrium after about 4 days, whereas Centreville's total gas phase continues to rise approaching equilibrium at a slower pace.

Looking at the total of the organics (right column of Figure 2), it is not surprising that the very polluted Beijing has the highest concentration of total organics while the cleanest location, Hyytiälä, has the lowest;





what is interesting, however, is that organics at these locations share similar volatility distributions. By the end of the 10-day simulations in the new scheme, the volatility distributions in Beijing and Hyytiälä are very similar to the emission factors distribution among the volatility, with factor differences of less than 0.1. This behavior is, again, a result of the low ˙OH concentrations in the two locations, and the low oxidation rate that limits the change in volatility distribution. Volatility is also temperature dependent, which is also relevant to the total aerosols present. In Beijing, we would expect more gas phase due to the higher temperatures. However, the larger amount of aerosols moves the partitioning point towards the aerosol phase, which offsets the temperature difference in the colder Hyytiälä case, and gives us similar results.

On the other hand, the volatility distributions in Centreville and Mexico City are very different from the applied emission factor distribution, except the two bins in the lowest volatility range M2 and M1. Due to the high concentrations of ˙OH, both sites have low gas phase organics concentrations because the high volatility gases are more efficiently oxidized and their less-volatile products partition into the aerosol phase. Therefore, the relative amount of organics from the high volatility range no longer resembles the applied emission factors. The organics in the high volatility range from P6 to P4, are totaling a factor of around 0.15, which is in sharp contrast to the factors of 0.4, 0.5, 0.8 applied to each of the respective bins.

### 5.2 Summer

The total mass concentration of organics in Beijing and Mexico City at the end of 10 days in July are about 130 μg/m$^3$ and 67 μg/m$^3$, very similar to the amounts in January. However, Centreville and Hyytiälä have much more organics than they did in January, with 90 μg/m$^3$ and 43 μg/m$^3$. The volatility distributions for the four locations in July (Figure 3) is also very different from that of January. Organics are all very low in the high and intermediate volatility ranges and are all high in the low volatility ranges, with less than 10% of the total organics in the gas phase in all four locations. This behavior means that at all locations oxidation is very strong, stronger than any place during January. This sharp change in behavior is caused by the difference in ˙OH concentrations during the two months. July's concentrations are much higher than those in January because ˙OH production is increased due to increased photolysis in the summer. The mean ˙OH concentration is about $1.5*10^7$ molecules/cm$^3$ in Beijing and Hyytiälä, and it is about $1*10^7$ molecules/cm$^3$ in Centreville and $2*10^7$ molecules/cm$^3$ in Mexico City. More ˙OH leads to faster oxidation of the gas phase organics and the consequent partitioning of the less-volatile oxidation products into the aerosol phase. This is evident in Figure 3, where the gas phase concentrations in all four locations are very low. In all cases, dynamic equilibrium was reached after just two days. They also exhibit a strong diurnal variability, as expected from the fast ˙OH oxidation, which decreases with decreasing volatility.



### 5.3 Mixing state

The temporal evolution of total organic aerosol mass concentration per population is shown in Figure 4 (absolute amounts) and Figure 5 (relative amounts). The first and third columns are results from the new scheme with condensing and coagulating organics for January and July, respectively, while the second and
fourth columns are results from the old scheme with only coagulating organics from January and July, respectively. The organic aerosol mass concentrations in Figure 4 correspond to the aerosol phase concentrations in Figures 2 and 3 (middle column), except they are now separated by population, whereas in the earlier two figures they were separated by mass tracers representing volatility. At a first glance, the population with the highest organic mass is BOC for January and July in both schemes. BOC is the population
that contains OC, BC, and sulfate, and is the end result of coagulation of all populations in our idealized cases. However, in the old scheme, populations OCC and OCS also have significant amounts of organics. This is because in the new scheme the emitted populations are ACC, BC1 and OCC, and organics that are emitted in the OCC population can condense on and/or coagulate with other populations, including being lost by evaporation and then repartitioning to other populations. Thus there is an additional loss mechanism of
organics from those populations in the new scheme. In addition, there is competition between the ACC and BC1 populations in both schemes, and in the new scheme, aerosol phase organics in the OCC population could either coagulate with the ACC population to form OCS, or they could coagulate with the BC1 population to form BOC. This competition determines how much OCS and BOC are formed, and it effects how much gas phase organics from the OCC population could condense on the two populations and the
distribution of organics among the populations. Since partitioning adds a loss mechanism to OCC, part of the evaporated mass will go to BOC, making it larger, and a more efficient scavenger of other particles. As a result, most organics coagulate with and condense on the BOC population and/or the OCS population, and together with the emitted OCC population, hold the most organics and dominate the mass fractions.

There is some similarity between the January and July results between the new and the old schemes (Figure 5).
This similarity means that the distribution of organics among aerosol populations is not significantly affected by season. This is consistent with a study by Bauer et al. (2013), where they found that the mixing state distribution is rather a characteristic of a region and not so much of a season, although the total (absolute) amounts by season may vary. By the end of the simulations, most locations have more organics present in the BOC population, except those in Centreville. The reason of this is sulfate; from the sulfate and black carbon
emissions listed in Table 3, we can calculate the sulfate to black carbon ratio in Centreville to be 2:1, higher than the corresponding ratios in all other locations. This high ratio helps the ACC population to survive the competition against BC1 for coagulation with OCC. This leads to higher OCS formation, which is available for gas phase organics to condense on, thus coagulation and condensation both bring more organics in the



OCS population during the first half of the simulation. These results show that the sulfate to black carbon ratio is important for the mixing state by delaying the inevitable BOC domination. Also, comparing the distribution fraction in Figure 5, volatile organics create rather different mixing states as those created by coagulation alone in the original scheme, meaning that the semi-volatility did alter mixing state significantly.

## 5   5.4 Size distribution

Another important factor on the evolution of aerosols is their size distribution. Shown in Figures 6 and 7 are the January size distributions from Mexico City and Centreville. The first row shows number concentration, the second row surface area, and the third row volume. The first two columns are results from the new and old schemes after 24 hours of simulation, and the right two columns are after 120 hours. The total number

concentration, surface area and volume from the eight populations are shown as dotted lines. Note also that these plots show the total aerosol size distribution per population, which includes the contribution of species other than organics.

The size distributions in July are very similar to January in all locations, therefore only January is shown here. Beijing, Hyytiälä and Mexico City exhibit somewhat similar size distributions (with different absolute

amounts), just as their mass fractions. The size distribution is dominated by OCC, OCS and BOC in the first 3 to 4 days, but later only by BOC. On the other hand, Centreville, similar to its mixing state, is different in size distribution of different aerosol populations from the other three locations. Therefore, only size distributions of Mexico City and Centreville are shown here.

In the new scheme for Mexico City after 24 hours of simulation, the number concentration has two modes.

OCC has even smaller size as Aitken mode sulfate AKK does, as a result of the evaporation of organics, but its number concentration is higher. OCS and BOC have started to form from coagulation of OCC with ACC and BC1, and their diameter, number concentration, surface area and volume are very similar, almost overlapping, with BOC slightly smaller in diameter. After 120 hours of simulation, OCC's number concentration has decreased significantly, from $2*10^6$ m$^{-3}$ to $0.5*10^6$ m$^{-3}$. This is because OCC is semi-volatile,

it has evaporated and condensed onto other populations, and at the same time its loss due to coagulation with other populations has increased, due to the increase of their number concentration and decrease in size. OCS size grew very slightly, but BOC grew significantly, with peaks of surface area and volume both increasing about one order of magnitude. Its peak surface area increased from $3*10^5$ μm$^2$m$^{-3}$ to $3*10^6$ μm$^2$m$^{-3}$, and its peak volume grew from about $1*10^5$ μm$^3$m$^{-3}$ to $1*10^6$ μm$^3$m$^{-3}$. BOC's growing large surface area is another

reason why it has so much organics and dominates the mass concentration: the greater the surface area, the more gas phase species are able to condense. This matches the mixing state results (Figure 5), where we saw after 24 hours ACC, OCC and BOC have high mass fractions, whereas after 120 hours OCC and OCS are negligible, and more than 90% of the total organic aerosol mass is in the BOC population.



In the old scheme, after 24 hours OCC has much higher number concentration (peaking at $1.2*10^7$ m$^{-3}$) and size than in the new scheme, and higher surface area and volume, due to its greater number and diameter. OCS and BOC are both fewer in number (peaks are $2*10^6$ m$^{-3}$ and $1*10^6$ m$^{-3}$ lower in the old scheme) but slightly greater in diameter than they are in the new scheme. Later, after 120 hours, OCC decreases in number to a

peak at $1.2*10^7$ m$^{-3}$, due to coagulation with ACC and BC1 to form more OCS and BOC. Therefore, OCS and BOC increased in number and size, with BOC seeing greater growth (the peak of number concentration increased 6-fold, from $3*10^6$ m$^{-3}$ to $1.8*10^7$ m$^{-3}$, the peak of surface area increased from $3*10^5$ μm$^2$m$^{-3}$ to $4.3*10^6$ μm$^2$m$^{-3}$ and the peak of volume increased from $1*10^5$ μm$^3$m$^{-3}$ to $1.6*10^6$ μm$^2$m$^{-3}$). For OCS we calculated more modest changes, with number, surface area and volume concentration peaks all increasing by

less than 50%: the number concentration from $2*10^6$ m$^{-3}$ to $3*10^6$ m$^{-3}$, surface area from $3*10^5$ μm$^2$m$^{-3}$ to $4*10^5$ μm$^2$m$^{-3}$ and volume from $0.8*10^5$ μm$^3$m$^{-3}$ to $1*10^5$ μm$^2$m$^{-3}$, as seen in the new scheme as well. However, BOC's growth in the old scheme is even greater than that in the new scheme. This slightly accelerated growth slows down at later hours (not shown), because BOC dominates faster in the new scheme than in the old one (Figure 4).

The Centreville size distributions tells a different story. In the early stages with the new scheme, OCS has greater number concentration and size than BOC does; OCS's peak number concentration is $2*10^6$ m$^{-3}$, more than double than that of BOC, while its surface area and volume are $2*10^6$ μm$^2$m$^{-3}$ and $0.5*10^5$ μm$^3$m$^{-3}$, whereas those of BOC are negligible. Later, OCS still outgrows BOC in number, but not in size. BOC shifts to greater diameters, therefore it has greater surface area and volume than OCS does after 120 hours. As for the

old scheme, OCC does not decrease in number from 24 hours to 120 hours as it does in Mexico City, but its number increases from $4*10^6$ m$^{-3}$ to $7*10^6$ m$^{-3}$. This means that in that period of time coagulation loss is less than the amount of OCC emitted, which is what was also seen earlier for the mass concentrations (Figure 2). At 120 hours, OCS has again higher number concentration than BOC does, but only slightly (peak number concentration difference is less than $1*10^6$ m$^{-3}$) and not as much as in the case of Mexico City, and the latter's

surface area and volume continue to be greater than those of the former due to its increasing diameter.

## 6 Conclusions

Organic aerosol volatility calculations were implemented into a new aerosol microphysics scheme, MATRIX-VBS. Results from idealized cases in Beijing, Centreville, Hyytiälä and Mexico City during summer and winter using the new scheme were compared against the original scheme and showed how the inclusion of

semi-volatility of organics affected aerosol mass concentration, as well as their mixing state and size distribution. Emission factors, ˙OH oxidation, temperature and total aerosol levels are the key factors determining organics' volatility distribution and mass concentration. The mixing state is affected by particle size and concentration, which determines coagulation and condensation pathways. Results from the new



scheme showed different mixing state distribution from the original scheme.

Going forward, the new scheme will be simplified, and we will reduce the number of tracers needed, in order to simplify the model and save computational resources, without losing the essential information needed for volatility. The simplified version of the box model will then be implemented in the NASA GISS ModelE

5 Earth System Model. While this study is purely theoretical, we will evaluate MATRIX-VBS after its implementation into GISS ModelE. We will gain even better understanding of how semi-volatile organics are altering aerosol mixing state, how meteorological conditions and pollution levels influence organics' volatility distribution, as well as their mixing state in the real world, and what implications these processes have on the climate system.

**Code Availability**

This model development is part of GISS ModelE Earth System Model, which is publicly available.

**Acknowledgements**

15 We thank the NASA Modeling, Analysis, and Prediction program, which supports the GISS ModelE development.



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



1    Table 1. Naming convention and parameters used in the VBS implementation described here.

| Parameter | 9 Virtual VBS Species | | | | | | | | |
|---|---|---|---|---|---|---|---|---|---|
| C* µg m$^{-3}$ at 298K | $10^{-2}$ | $10^{-1}$ | $10^{0}$ | $10^{1}$ | $10^{2}$ | $10^{3}$ | $10^{4}$ | $10^{5}$ | $10^{6}$ |
| log(C*) | -2 | -1 | 0 | 1 | 2 | 3 | 4 | 5 | 6 |
| Gas phase name[1] | VBSm2g | VBSm1g | VBSm0g | VBSp1g | VBSp2g | VBSp3g | VBSp4g | VBSp5g | VBSp6g |
| Aerosol tracer name in each MATRIX population | OCM2 | OCM1 | OCM0 | OCP1 | OCP2 | OCP3 | OCP4 | OCP5 | OCP6 |
| Mass-based emission factors applied to POA emissions (Shrivastava et al., 2008) | 0.03 | 0.06 | 0.09 | 0.14 | 0.18 | 0.30 | 0.40 | 0.50 | 0.80 |

3    [1]: in VBSxnp, x is m for minus and p for plus, n is the absolute value of log(C*), and p is g for the gas phase and a for the sum across all
4    populations of the aerosol-phase concentration.



Table 2. Aerosol population chemical composition in MATRIX.

| Population abbreviation | Description | Composition (constituents other than $NH^+_4$, $NO^-_3$, and $H_2O$) |
|---|---|---|
| AKK | sulfate (Aitken mode) | $SO_4^{2-}$ |
| ACC | sulfate (accumulation mode) | $SO_4^{2-}$ |
| OCC | organic carbon | OC, $SO_4^{2-}$ |
| BC1 | fresh black carbon (<5% coating) | BC, $SO_4^{2-}$ |
| BC2 | aged (by condensation) black carbon (>5% coating) | BC, $SO_4^{2-}$ |
| BCS | aged (by coagulation) black carbon | BC, $SO_4^{2-}$ |
| BOC | black and organic carbon | BC, OC, $SO_4^{2-}$ |
| OCS | organic carbon and sulfate | OC, $SO_4^{2-}$ |
| SSA | sea salt (accumulation mode) | sea salt, $SO_4^{2-}$ |
| SSC | sea salt (coarse mode) | sea salt, $SO_4^{2-}$ |
| DD1 | dust (accumulation mode; <5% coating) | mineral dust, $SO_4^{2-}$ |
| DD2 | dust (coarse mode; <5% coating) | mineral dust, $SO_4^{2-}$ |
| DS1 | dust (accumulation mode; >5% coating ) | mineral dust, $SO_4^{2-}$ |
| DS2 | dust (coarse mode; >5% coating) | mineral dust, $SO_4^{2-}$ |
| MXX | mixed (all components) | BC,OC, mineral dust, sea salt, $SO_4^{2-}$ |





Table 3. Conditions of each location used in the simulations, taken from the GISS ModelE for January and July 2006.

| January 2006 | | Units | Beijing | Centreville | Hyytiälä | Mexico City |
|---|---|---|---|---|---|---|
| **Parameters** | Temperature | K | 270 | 279 | 260 | 283 |
| | Pressure | hPa | 1007 | 996 | 1009 | 797 |
| | RH | % | 46.8 | 77.7 | 79.5 | 62.5 |
| **Gaseous emissions** | NO$_x$ | | 216.5 | 92.4 | 169.7 | 148.7 |
| | CO | | 6943.3 | 1199.3 | 557.3 | 2308.4 |
| | Alkenes | | 4.3 | 0.3 | 0.1 | 1.3 |
| | Paraffin | pptv/hr | 8.2 | 2.1 | 0.6 | 10.5 |
| | Terpenes | | 1.8 | 26.3 | 9.4 | 25.8 |
| | Isoprene | | 1.3 | 23.8 | 0.0 | 0.0 |
| | SO$_2$ | | 555.8 | 191.7 | 24.1 | 538.7 |
| | NH$_3$ | | 181.3 | 24.2 | 50.7 | 63.3 |
| **Aerosol emissions** | sulfate | | 0.06 | 0.02 | 0.003 | 0.05 |
| | black carbon | µg/m$^3$/hr | 0.09 | 0.01 | 0.008 | 0.03 |
| | organics* | | 0.19 | 0.03 | 0.02 | 0.11 |

| July 2006 | | Units | Beijing | Centreville | Hyytiälä | Mexico City |
|---|---|---|---|---|---|---|
| **Parameters** | Temperature | K | 304 | 303 | 292 | 289 |
| | Pressure | hPa | 986 | 995 | 998 | 800 |
| | RH | % | 59.8 | 61.8 | 77.8 | 83.1 |
| **Gaseous emissions** | NO$_x$ | | 281.3 | 124.3 | 200.9 | 165.3 |
| | CO | | 8111.9 | 1749.9 | 630.5 | 2276.1 |
| | Alkenes | | 5.0 | 0.5 | 0.1 | 1.3 |
| | Paraffin | pptv/hr | 9.6 | 2.7 | 0.7 | 10.7 |
| | Terpenes | | 36.9 | 145.4 | 87.6 | 44.9 |
| | Isoprene | | 916.1 | 795.5 | 47.2 | 0.0 |
| | SO$_2$ | | 653.7 | 206.5 | 26.8 | 549.5 |
| | NH$_3$ | | 211.7 | 38.7 | 58.1 | 63.3 |
| **Aerosol emissions** | sulfate | | 0.06 | 0.02 | 0.002 | 0.05 |
| | black carbon | µg/m$^3$/hr | 0.10 | 0.01 | 0.01 | 0.03 |
| | organics* | | 0.21 | 0.03 | 0.07 | 0.11 |



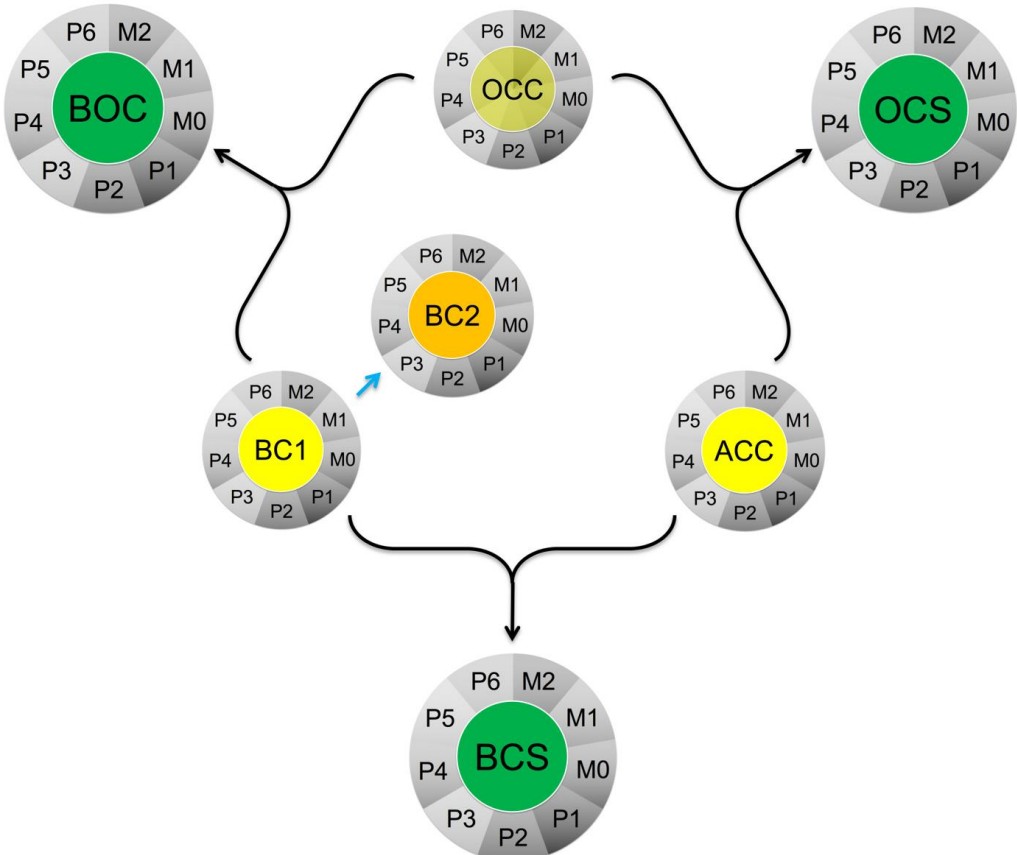

Figure 1. Schematic showing coagulation pathways among organic-containing aerosol populations as colored circles, with 9 organic VBS species condensed as grey outer circles. In yellow are the emitted donor aerosol populations, and green are the mixed recipient populations. OCC has a semi-transparent yellow core because it is actually emitted as the VBS species that can serve as condensation medium for gaseous VBS species, represented by the grey outer circles. In orange is population BC2, which contains >5% coating of sulfate and organics, which is formed rapidly from the growth of population BC1, which has <5% sulfate/organics coating.





Figure 2. Temporal evolution of the mass concentration of semi-volatile organics in the gas phase (left column), aerosol phase (across all populations; middle column) and total (right column) using the new scheme for January. The total of non-volatile organics from the original scheme is shown in black dash-dotted lines in the right column.







Figure 3. Same as Figure 2, for July.





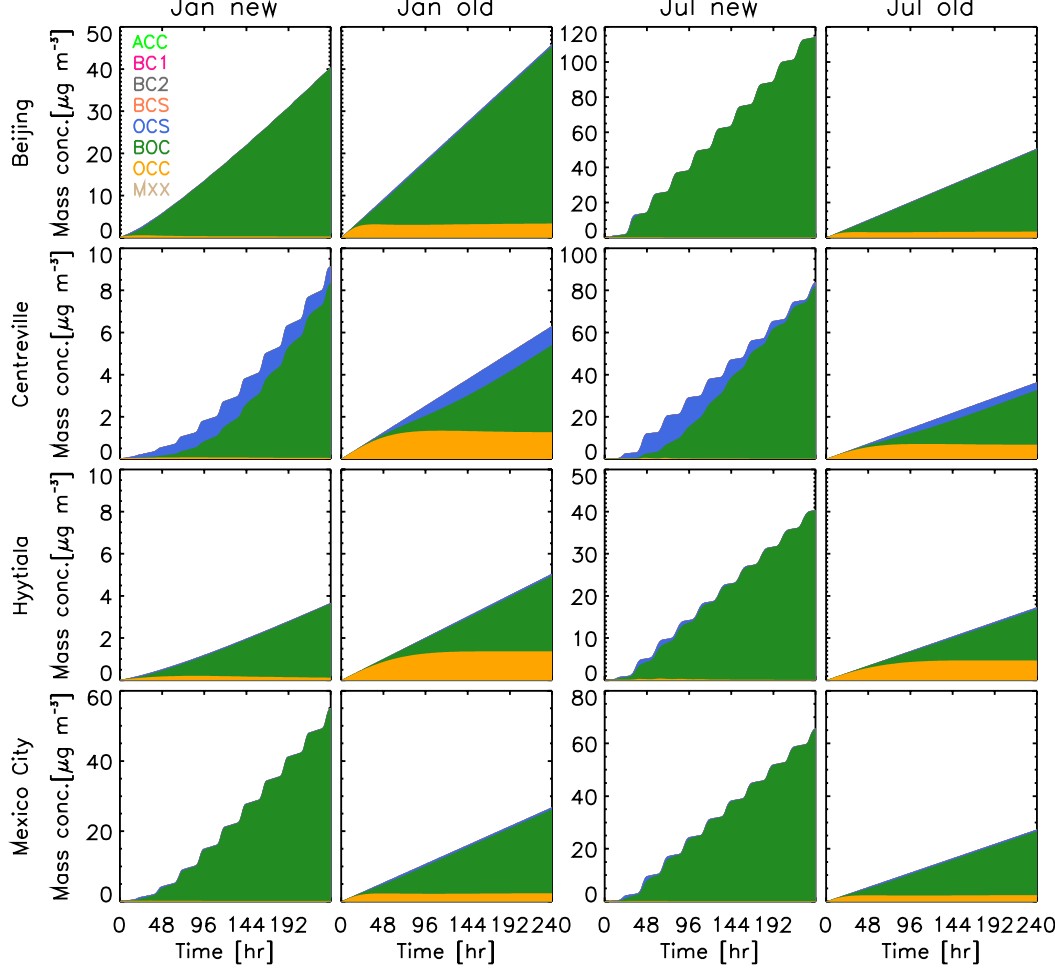

Figure 4. Temporal evolution of organic aerosol mass concentration in each organics-containing population from the new scheme (left two columns, first column for January, second column for July), and the old scheme (right two columns, third column for January, fourth column for July).



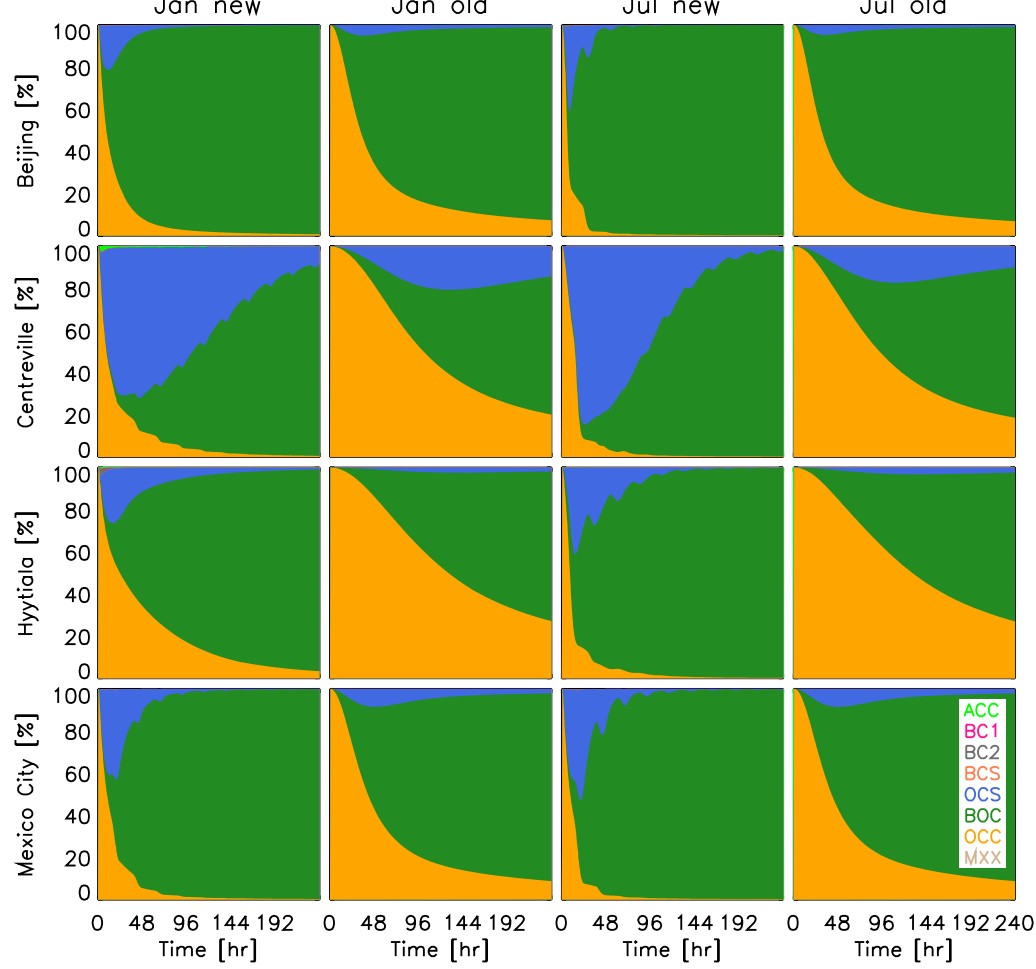

Figure 5. Temporal evolution of organic aerosol mass concentration fraction in each organics-containing
population from the new scheme (left two columns, first column for January, second column for July), and the
5   old scheme (right two columns, third column for January, fourth column for July).





Figure 6. Organics-containing aerosol populations (except MXX) and AKK (Aitken mode sulfate) size distributions for Mexico City in January. Top row: number concentration, middle row: surface area, bottom row: volume. Total of all populations in dotted black lines.





Figure 7. Same as Figure 6 for Centreville.

