# Peer review of "MATRIX-VBS (v1.0): an aerosol microphysics model including organic aerosol volatility"

_Geoscientific Model Development, 2016_

## Short Comment (SC1) · 15 Sep 2016

Dear authors,

In my role as Executive editor of GMD, I would like to bring to your attention our Editorial version 1.1:

http://www.geosci-model-dev.net/8/3487/2015/gmd-8-3487-2015.html

This highlights some requirements of papers published in GMD, which is also available on the GMD website in the 'Manuscript Types' section:

http://www.geoscientific-model-development.net/submission/manuscript_types.html

[Figure]

In particular, please note that for your paper, the following requirement has not been met in the Discussions paper:

- "The main paper must give the model name and version number (or other unique identifier) in the title."

Please add a version number for MATRIX-VBS in the title upon your revised submission to GMD.

Concerning your statements in the Code Availability Section, I like to inquire, if the box model, which is subject to your publication, might also be available independent of the full GISS ModelE Earth System Model.

Yours,

Astrid Kerkweg

---

## Referee Comment (RC1) · Anonymous Referee #1 · 29 Sep 2016

This paper documents the addition of the VBS to a box model of MATRIX and shows results from some basic test simulations. It generally fits with GMD and overall seems scientifically sound. However, the paper is missing much information that would be necessary for someone to repeat the analysis, and I believe there is an issue with Figures 6 and 7. I support publication of this paper once my concerns are addressed.

Figures 6 and 7: In Figures 6 and 7, for each population, the median diameters of the number, surface-area, and volume distributions are all the same. For example, in Figure 6 for "T=120hr new", the median diameter for BOC is 80 nm in each of the three distributions. The median diameter of a mode is only the same for number, surface area, and volume distributions if the particles are in that mode are all exactly

the same size. Yet, the modes representing the populations in Figures 6 and 7 are shown having a finite width, which means that the median diameter of the surface-area median diameter should be larger than the number-median diameter, and the volume-median diameter larger than that for surface area. I'm guessing that MATRIX is only simulating 2 moments per population (though this is never explicitly stated) and an assumption is made about a 3rd moment is made in order to get the modal width in Figures 6 and 7. This 3rd-moment assumption is fine, but this 3rd-moment assumption needs to be consistantly used such that the median diameters shift between the three distributions.

The following information is either missing from the paper or shows up later than ideal:

Enthalpy of vaporization that drives the temperature dependence of C*s. How many moments tracked per population? I'm guessing 2 since that what I remember from previous MATRIX papers, but it needs to be explicitly stated here. Which moments? Number and mass (volume)? In Figures 6 and 7, it looks like the moments are converted to modes. If there are just 2 moments, I'm guessing you assumed a fixed width (this looks to be the case). What width did you use or what did you assume about a 3rd moment? Also see comment about the number, surface area, and volume median diameters of the modes incorrectly being the same. Several things should be explicitly stated in the methods but aren't discussed until later in the paper: (a) the duration of the simulations (10 days), (b) are emission continuous?, (c) oxidant concentrations. What is the diurnal cycle of temperature and oxidant concentrations? What are the gas-phase chemical rate constants? Is condensation/evaporation to each population calculated through kinetic mass transfer, or are the populations and gas-phase assumed to be in instantaneous equillibrium? Instantaneous equillibrium might work ok here where all of the populations interacting with the gas-phase organics are essentially all accumulation mode (and thus the populations have similar equillibrium timescales); however, this assumption will likely fail when the authors begin to consider organic uptake to the nucleation and Aitken modes, and the equillibration time will vary between modes.

Other comments

Title: "an evolving organic aerosol volatility" sounds awkward to me. What about "an organic aerosol volatility scheme" or removing the work "an" from the current title?

P1 L21: Can you define the low- and high-volatility ranges?

P1 L22: "The *final* volatility distribution..."?

P4 L27: This sentence is confusing. I thought that in the old scheme, non-volatile SOA condensed onto modes too (see P3 L21), but this sentence says coagulation is the only process involving organics.

Figures 2 and 3: I think it would make more sense to have the black line showing the mass in the old scheme in the "aerosol phase" panels rather than the "total" panels since the old scheme was only tracking the aerosol mass. I view the aerosol mass in the new scheme and the old scheme's aerosol mass as the apples-to-apples comparison.

P7 L30: "...total gas-phase *concentration* reaches..."

P8 L6: "...would expect *higher* gas-phase *concentrations* due to..."

Copy editing: The paper would benefit from having compound adjectives being properly hyphenated. Also commas appear in some places they shouldn't and don't appear in some places they should.

---

## Referee Comment (RC2) · Anonymous Referee #2 · 17 Oct 2016

General Comments:

The paper describes the implementation of a volatility basis set (VBS) approach within a box model version of the Multiconfiguration Aerosol TRacker of mIXing state (MA-TRIX) scheme.

The paper is generally well written but would benefit from careful proof reading to ensure that everything is written as concisely and accurately as possible. Whilst the paper fits within the scope of GMD, more detail would be needed for anyone to reproduce this study - I would recommend publication once the below issues have been resolved.

Specific Comments:

[Figure]

- Page 3, line 3-6: This sentence is talking about the role of semi-volatile organics but mentions new particle formation - I would suggest moving the reference to new particle formation to the previous sentence about very low volatility organics, and including some additional references, i.e.: "It has been established that highly oxidised, low-volatility organics play a key role in new particle formation (Metzger et al., 2010; Riccobono et al., 2014; Kirkby et al., 2016) and particle growth (Trostl et al., 2016), while the range...." [this is alluded to in your Methods section but not clear here]

- Page 4, line 1-14: There are a few more details that would be useful to include in this model description. Specifically: Clarify whether this model tracks both aerosol number and aerosol mass? [I assume both from Figures 6 and 7] What are the "aerosol populations" – they seem to be specific modes of particular composition (perhaps refer the reader to Table 2 at this point)? What are the diameter ranges of these populations? (this also becomes relevant when interpreting Figures 6 and 7)

- Page 5, lines 25 - 30: Can you expand on what these mass-based emission factors represent? (enough for the reader to understand without having to refer to Shrivastava et al., 2008) To what is the additional factor of 1.5 applied?

- While the VBS scheme is outlined in the Model Description, it would be useful to include a brief description of how this influences, and what controls, the gas-particle partitioning among the different populations.

- Reaction with OH is mentioned as a mechanism for chemical aging of the organic species, is the OH concentration taken from GISS ModelE? How often are the concentrations updated? How does this process occur (does it occur at a specified reaction rate for each of the volatility bins of organic species?)

- Where do the products of the oxidation of biogenic organics go? Into the lowest volatility bin? Or are they treated entirely separately?

- Some of the information in Section 4 seems like it would be more appropriate in the

model (i.e., description of the treatment of BC) / VBS description (i.e., description of the emission rates for the VBS species), rather than the discussion of the simulations?

- Page 6, lines 19 – 20: this currently sounds like each emission factor is 2.5 (which is not the case?), perhaps rephrase this to clarify what you mean

- In a couple of places, compounds that are low-, intermediate-, and high-volatility are referred to (e.g., Page 7, lines 13-15); in terms of the volatility bins used here, how are these categories defined?

- Page 22, Figure 2: Refer to Table 1 or Figure 1 to explain the legend entries and define OCAR Page 24/25, Figure 4/5: The layout of the columns doesn't quite match the description in the caption

Technical Suggestions:

I have a number of technical suggestions that may help improve the readability of the manuscript.

General comments:

On several occasions, quantities are referred to as "about" ##, it would be better to either refer to a more specific value or refer to this same value as "approximate".

Specific suggestions:

Page 1, line 21: should this be "high volatility" rather than "high volatile"?

Page 2, line 21: insert "the" between "with" and "hydroxyl"

Page 2, line 27: remove "it" from between "space" and "also"

Page 2, line 28/29: specify that the "it" being referred to here is the 2D-VBS?

Page 3, line 23: replace "terpenes emissions" with "emissions of monoterpenes" [if you are only referring to monoterpenes here]

Page 4, line 21: remove the word "ones" from between "volatile" and "and"

Page 5, line 11: insert a hyphen into "nonvolatile"

Page 5, line 12-14: rephrase this sentence, it is currently not clear what you mean – and be consistent with writing numerical values as words or numbers, i.e. "eight" v. "8"

Page 5, line 18: you could replace "of different" with "and" ?

Page 5, line 19: you could move "(Mexico City) to the end of this description (after "tropics") to avoid confusion

Page 5, line 23: you could separate this into two sentences, with the first ending after "Table 3" and the second beginning with "Here we do not include….."

Page 6, line 30: replace "least one" with "least volatile"

Page 7, line 11: should this say "more oxidized" rather than "less oxidized"?

Page 7, line 18: replace "same with that" as "same as that"

Page 8, line 19: replace "much more" with "higher concentrations"

Page 8, line 20: replace "is also very different" with "are also very different"

Page 9, line 18: replace "effects" with "affects"

Page 9, line 29: replace "reason of this" with "reason for this"

References:

Kirkby J; Duplissy J; Sengupta K; Frege C; Gordon H; Williamson C; Heinritzi M; Simon M; Yan C; Almeida J; Trostl J; Nieminen T; Ortega IK; Wagner R; Adamov A; Amorim A; Bernhammer AK; Bianchi F; Breitenlechner M; Brilke S; Chen X; Craven J; Dias A; Ehrhart S; Flagan RC; Franchin A; Fuchs C; Guida R; Hakala J; Hoyle CR; Jokinen T; Junninen H; Kangasluoma J; Kim J; Krapf M; Kurten A; Laaksonen A; Lehtipalo K; Makhmutov V; Mathot S; Molteni U; Onnela A; Perakyla O; Piel F; Petaja T; Praplan

AP; Pringle K; Rap A; Richards NAD; Riipinen I; Rissanen MP; Rondo L; Sarnela N; Schobesberger S; Scott CE; Seinfeld JH; Sipila M; Steiner G; Stozhkov Y; Stratmann F; Tomé A; Virtanen A; Vogel AL; Wagner AC; Wagner PE; Weingartner E; Wimmer D; Winkler PM; Ye P; Zhang X; Hansel A; Dommen J; Donahue NM; Worsnop DR; Baltensperger U; Kulmala M (2016) Ion-induced nucleation of pure biogenic particles, Nature, 533, pp.521-526. doi: 10.1038/nature17953

Pierce, J. R., Riipinen, I., Kulmala, M., Ehn, M., Petäjä, T., Junninen, H., Worsnop, D. R., and Donahue, N. M.: Quantification of the volatility of secondary organic compounds in ultrafine particles during nucleation events, Atmos. Chem. Phys., 11, 9019-9036, doi: 10.5194/acp-11-9019-2011, 2011.

Riccobono F; Schobesberger S; Scott CE; Dommen J; Ortega IK; Rondo L; Almeida J; Amorim A; Bianchi F; Breitenlechner M; David A; Downard A; Dunne EM; Duplissy J; Ehrhart S; Flagan RC; Franchin A; Hansel A; Junninen H; Kajos M; Keskinen H; Kupc A; Kürten A; Kvashin AN; Laaksonen A; Lehtipalo K; Makhmutov V; Mathot S; Nieminen T; Onnela A; Petäjä T; Praplan AP; Santos FD; Schallhart S; Seinfeld JH; Sipilä M; Spracklen DV; Stozhkov Y; Stratmann F; Tomé A; Tsagkogeorgas G; Vaattovaara P; Viisanen Y; Vrtala A; Wagner PE; Weingartner E; Wex H; Wimmer D; Carslaw KS; Curtius J; Donahue NM; Kirkby J; Kulmala M; Worsnop DR; Baltensperger U (2014) Oxidation products of biogenic emissions contribute to nucleation of atmospheric particles, Science, 344, pp.717-721. doi: 10.1126/science.1243527

Tröstl J; Chuang WK; Gordon H; Heinritzi M; Yan C; Molteni U; Ahlm L; Frege C; Bianchi F; Wagner R; Simon M; Lehtipalo K; Williamson C; Craven JS; Duplissy J; Adamov A; Almeida J; Bernhammer A-K; Breitenlechner M; Brilke S; Dias A; Ehrhart S; Flagan RC; Franchin A; Fuchs C; Guida R; Gysel M; Hansel A; Hoyle CR; Jokinen T; Junninen H; Kangasluoma J; Keskinen H; Kim J; Krapf M; Kuerten A; Laaksonen A; Lawler M; Leiminger M; Mathot S; Moehler O; Nieminen T; Onnela A; Petaejae T; Piel FM; Miettinen P; Rissanen MP; Rondo L; Sarnela N; Schobesberger S; Sengupta K; Sipilae M; Smith JN; Steiner G; Tome A; Virtanen A; Wagner AC; Weingartner E; Wimmer D;

Winkler PM; Ye P; Carslaw KS; Curtius J; Dommen J; Kirkby J; Kulmala M; Riipinen I; Worsnop DR; Donahue NM; Baltensperger U (2016) The role of low-volatility organic compounds in initial particle growth in the atmosphere, Nature, 533, pp.527-531. doi: 10.1038/nature18271

———————————————————

---

## Referee Comment (RC3) · Anonymous Referee #3 · 24 Oct 2016

In this paper, Gao et al. updated the MATRIX aerosol microphysical scheme to include the volatility basis set (VBS) framework for simulating the organic aerosol formation and growth. The manuscript is of interest to the GMD audience and the scientific methodology used sounds valid. Furthermore, the presented module has the potential to be a great addition to the NASA GISS ModelE ESM. However, I would recommend to describe the new scheme in more detail prior to publication.

**Specific comments:**

1. Page 3 lines 21-25: This paragraph needs to be moved to the model description

2. Section 2.1: I suggest to describe more in detail here the aerosol populations used by the MATRIX model.

3. Section 2.2: I suggest to improve the description of the VBS framework used. Important information that affect the presented results are missing here: For instance, the description of the ageing mechanism (i.e., the oxidation rate constants used, the reduction in volatility after each oxidation step, and the oxygen mass added per oxidation). Furthermore, how do you describe the formation of SOA from traditional VOCs in the model? Do you use a narrower volatility distribution (e.g., up to $10^3$)? Do you use aerosol yields for the oxidation of VOCs (if so please report them)? Another issue is how do you perform the partition between the two phases. Do you assume instant equilibrium? Do you account for the temperature dependence of saturation concentrations? If so, what are the enthalpies of vaporization used for each of your organic species? Finally, since you use terms such as "high volatility range", "intermediate volatility range", and "low volatility range" in your results I would recommend to define these terms here in respect to the effective saturation concentration.

4. Page 5, lines 1-4: Which is the extra population compared to the "14 populations" configuration? Furthermore, according to Table 2, the aerosol populations ACC, BC1, BC2, and BCS do not contain organics. Please explain what do you mean that these populations "are set to contain organics as semi-volatile VBS species".

5. Page 6 lines 1-2: Why the overestimation of biomass burning emission factors is not an issue for your experiments? Your experiments include forested areas during summer (where you have high emissions from open biomass burning) and highly populated areas during winter (where you have high biomass burning emissions from residential heating).

6. Page 7 lines 1-12: Most of this discussion belongs to the model description.

7. Page 11 line 30: You did not only include the volatility of the organics, but also their reactivity (by ageing with OH).

8. Table 2: What is the size of the mode (Aitken, accumulation, or coarse) used for the aerosol populations OCC, BC1, BC2, BCS, BOC, OCS, and MXX? Also, does the OCC has sulfate? Because in that case it seems to be identical to the OCS.

9. Figure 2: "P2" and "OCAR" seem to have very similar color. Please change the color of "P2". Furthermore, it would be nice to add for comparison the "OCAR" dashed line in the "Aerosol phase" column as well.

10. Figure 4: The label is wrong. It writes that the second column is "July new" and the third for "January old" which is not the case.

---

## Author Comment (AC1) · 29 Dec 2016

Dear Executive Editor and Referees,

We received your comments and suggestions for our discussion paper "MATRIX-VBS: implementing organic aerosol volatility in an aerosol microphysics model" (now: "MATRIX-VBS (v1.0): an aerosol microphysics model including organic aerosol volatility") on GMDD. We thank you very much for your efforts in evaluating our submission.

We have used the comments received to guide our revision of the discussion paper. Almost all of the advice received were incorporated into the revised paper. Point-by-point answers to all questions raised are listed below, together with highlighted changes in the revised manuscript.

We hereby resubmit the revised discussion paper to be considered for publication in *Geoscientific Model Development*. We confirm that all authors listed on the manuscript concur with submission in its revised form. Should you have any remaining questions, we will be happy to address them.

Sincerely,

C. Y. Gao, K. Tsigaridis and S. E. Bauer

**Executive Editor**

**…In particular, please note that for your paper, the following requirement has not been met in the Discussions paper: • "The main paper must give the model name and version number (or other unique identifier) in the title." Please add a version number for MATRIX-VBS in the title upon your revised submission to GMD.**

Now the title reads: "MATRIX-VBS (v1.0): an aerosol microphysics model including organic aerosol volatility".

**Concerning your statements in the Code Availability Section, I like to inquire, if the box model, which is subject to your publication, might also be available independent of the full GISS ModelE Earth System Model.**

Yes, the box model is available on request. We added this information in the code availability section: "In addition, the box model used here is available on request".

1. **Figures 6 and 7: In Figures 6 and 7, for each population, the median diameters of the number, surface-area, and volume distributions are all the same. For example, in Figure 6 for "T=120hr new", the median diameter for BOC is 80 nm in each of the three distributions. The median diameter of a mode is only the same for number, surface area, and volume distributions if the particles are in that mode are all exactly the same size. Yet, the modes representing the populations in Figures 6 and 7 are shown having a finite width, which means that the median diameter of the surface-area median diameter should be larger than the number-median diameter, and the volume median diameter larger than that for surface area. I'm guessing that MATRIX is only simulating 2 moments per population (though this is never explicitly stated) and an assumption is made about a 3rd moment is made in order to get the modal width in Figures 6 and 7. This 3rd-moment assumption is fine, but this 3rd-moment assumption needs to be consistently used such that the median diameters shift between the three distributions.**
Thank you for pointing this out. We discovered a bug in the plotting of the size distribution which was not affecting the model results. Now Figures 6 and 7 have been updated and the discussion has been modified accordingly. In addition, it is true that MATRIX only simulates 2 moments per population, and each population has a fixed width. Although we never explicitly mention the term "moments" in the manuscript, the fact that we simulate number and mass concentrations per population is explicitly mentioned in the manuscript, including the abstract. However, to make this point even clearer, we expanded an explicit sentence about it in section 2.1: "MATRIX is designed to resolve the aerosol temporal evolution and represent the mixing states of a user-selected set of aerosol populations, which are modes of different composition as listed in Table 2, tracking two moments each, number and mass, while keeping the width of the distribution fixed".

2. **The following information is either missing from the paper or shows up later than ideal:**
2.1 **Enthalpy of vaporization that drives the temperature dependence of C*s.**
The enthalpy of vaporization is calculated using equation 12 from Epstein et al., 2010. This is now included in Table 1.

| bin | Enthalpy of vaporization |
|-----|--------------------------|
| -2  | 153 |
| -1  | 142 |
| 0   | 131 |
| 1   | 120 |
| 2   | 109 |
| 3   | 98  |
| 4   | 87  |
| 5   | 76  |
| 6   | 65  |

**2.2 How many moments tracked per population? I'm guessing 2 since that what I remember from previous MATRIX papers, but it needs to be explicitly stated here. Which moments? Number and mass (volume)?**

Yes, two moments, number and mass. See also answer to point 1.

**2.3 In Figures 6 and 7, it looks like the moments are converted to modes. If there are just 2 moments, I'm guessing you assumed a fixed width (this looks to be the case). What width did you use or what did you assume about a 3rd moment?**

We included the fixed widths for the modes at the bottom of Table 2: "*The sigma values for all populations are 1.80, except for AKK, which has a sigma of 1.60, and for SSC and MXX, which both have a sigma of 2.00."

**Also see comment about the number, surface area, and volume median diameters of the modes incorrectly being the same.**

Thank you, this was a plotting bug, please see answer to point 1.

**3. Several things should be explicitly stated in the methods but aren't discussed until later in the paper:**

**3.1 the duration of the simulations (10 days)**

This is already stated in section 4, description of the simulations. We changed "ten" to "10".

**3.2 are emission continuous?**

Yes. We added the word "continuously" in the emissions description in section 4. The sentence now reads: "Semi-volatile POA, sulfate in the accumulation mode, and black carbon, are emitted continuously in the OCC, ACC, BC1 populations, respectively, shown in Figure 1 as yellow circles."

**3.3 oxidant concentrations? What is the diurnal cycle of temperature and oxidant concentrations?**

We did not consider a diurnal cycle of temperature, for simplicity. Inserted in page 5 line 23: "All parameters and emissions are held constant throughout the simulations". We also changed the row titles in Table 3 to read "Fixed parameters". The oxidant concentrations change with time, as calculated by photochemistry and the diurnal variability of solar zenith angle which affects photolysis rates. The mean calculated OH radical concentrations are already mentioned in the results section, but we selected to not show a plot of them, for clarity. They are shown below.

[Figure]

**3.4 What are the gas-phase chemical rate constants?**

Gas phase organics are oxidized by OH radicals with a rate constant of $1*10^{-11}$ cm$^3$ s$^{-1}$ (Donahue et al., 2006). We added this information in section 2.2.

**3.5 Is condensation/evaporation to each population calculated through kinetic mass transfer, or are the populations and gas-phase assumed to be in instantaneous equillibrium? Instantaneous equillibrium might work ok here where all of the populations interacting with the gas-phase organics are essentially all accumulation mode (and thus the populations have similar equillibrium timescales); however, this assumption will likely fail when the authors begin to consider organic uptake to the nucleation and Aitken modes, and the equillibration time will vary between modes.**

Yes, we used instantaneous equilibrium, as described in Donahue et al. (2006), which is based on the Pankow (1994) theory. This is now mentioned in the revised manuscript in sections 2.2 and 3.3.

**4. Other comments Title: "an evolving organic aerosol volatility" sounds awkward to me. What about "an organic aerosol volatility scheme" or removing the work "an" from the current title?**

We modified the title to "MATRIX-VBS (v.1): an aerosol microphysical model including organic aerosol volatility".

**5. P1 L21: Can you define the low- and high-volatility ranges?**

We added our definition in the VBS framework description: "We classify organics as Murphy et al., 2014 does:  low-volatility organics are in bins $10^{-2}$ to $10^{-1}$ μg m$^{-3}$ (M2 and M1 in Table 1), semi-volatile organics are in bins $10^0$ to $10^2$ μg m$^{-3}$ (M0, P1, P2), and intermediate-volatility organics are in bins $10^3$ to $10^6$ μg m$^{-3}$ (P3, P4, P5, and P6). " We also modified the terms accordingly throughout the paper.

We also modified the abstract: "Emitted semi-volatile primary organic aerosols evaporate almost completely in the intermediate volatility range, while they remain in the particle phase in the low volatility range."

**6. P1 L22: "The \*final\* volatility distribution. . ."?**
At any point in time the volatility distribution would depend on those factors. We modified the sentence to read "The volatility distribution at any point in time depends on…"

**7. P4 L27: This sentence is confusing. I thought that in the old scheme, non-volatile SOA condensed onto modes too (see P3 L21), but this sentence says coagulation is the only process involving organics.**
Non-volatile SOA do not condense onto any modes (aerosol populations), they are directly emitted into those, same as other primary aerosols, e.g. BC, as already explicitly stated in the lines mentioned by the reviewer: "In the original version of the MATRIX model, organics only contribute to particle growth and mix with other aerosol species via coagulation. Primary organic aerosols are emitted only as non-volatile particulate organic matter, and do not exist in the gas phase or interact with other aerosol populations". Page 3 line 21 does not say non-volatile SOA condensed onto modes, it says "The semi-volatile nature of biogenic SOA is not represented in the VBS framework in this work", so we are not certain why the reviewer was confused here.

**8. Figures 2 and 3: I think it would make more sense to have the black line showing the mass in the old scheme in the "aerosol phase" panels rather than the "total" panels since the old scheme was only tracking the aerosol mass. I view the aerosol mass in the new scheme and the old scheme's aerosol mass as the apples-to-apples comparison.**
The reason why we decided to plot the original in the total rather than the aerosol phase is to directly visualize the Shrivastava et al. (2008) emission factors impact on the total organics, which takes into account IVOC emissions. We do agree though that for an apples-to-apples the line is probably better to be presented in the aerosol phase, which is what we now show.

**9. P7 L30: "...total gas-phase \*concentration\* reaches. . ."**
Corrected.

**10. P8 L6: "...would expect \*higher\* gas-phase \*concentrations\* due to. . ."**
Corrected.

**11. Copy editing: The paper would benefit from having compound adjectives being properly hyphenated. Also commas appear in some places they shouldn't and don't appear in some places they should.**

When gas-phase is a compound adjective it's hyphenated, when it's not, for instance "in the gas phase" then it's not hyphenated. Same applies for aerosol-phase. In addition, after acceptance for publication and during the production phase, there is a copy-editing process that should take care of any remaining issues on that front. Taken from the GMD website (http://www.geoscientific-model-development.net/production.html): "Where applicable, the article undergoes the copy-editing process (spelling, grammar, sentence structure) and the changes are incorporated into the text and again converted into a PDF."

**REFEREE #2**

1. **Page 3, line 3-6: This sentence is talking about the role of semi-volatile organics but mentions new particle formation - I would suggest moving the reference to new particle formation to the previous sentence about very low volatility organics, and including some additional references, i.e.: "It has been established that highly oxidised, low-volatility organics play a key role in new particle formation (Metzger et al., 2010; Riccobono et al., 2014; Kirkby et al., 2016) and particle growth (Trostl et al., 2016), while the range. . .."** [this is alluded to in your Methods section but not clear here]
   Now the sentence reads: "It has been established that the highly oxidized, very low volatility organics play a key role in particle formation (Metzger et al., 2010; Paasonen et al., 2013; Riccobono et al., 2014; Kirkby et al., 2016) and particle growth (Trostl et al., 2016),"

2. **Page 4, line 1-14: There are a few more details that would be useful to include in this model description. Specifically: Clarify whether this model tracks both aerosol number and aerosol mass? [I assume both from Figures 6 and 7] What are the "aerosol populations" – they seem to be specific modes of particular composition (perhaps refer the reader to Table 2 at this point)? What are the diameter ranges of these populations? (this also becomes relevant when interpreting Figures 6 and 7)**
   Please see response to referee #1 2.2, the two moments, number and mass, are clarified in page 3, line 8. The reference to Table 2 for the clarifying what aerosol populations are is included as well. Now the sentence reads: "MATRIX is designed to resolve the aerosol temporal evolution and represent the mixing states of a user-selected set of aerosol populations, which are modes of different composition as listed in Table 2, tracking two moments each, number and mass, while keeping the width of the distribution fixed."
   Populations have a characteristic size at emission time, and can grow or shrink to the whole size distribution range, contrary to other modal models where mass and number move from one mode to another. A notable exception relevant to the work presented here is the move of aerosol number and mass from BC1 to BC2; this, however, happens with compositional criteria, not size assumptions. For details, see Bauer et al., 2008.

3. **Page 5, lines 25 - 30: Can you expand on what these mass-based emission factors represent? (enough for the reader to understand without having to refer to Shrivastava et al., 2008) To what is the additional factor of 1.5 applied?**
   The emission factors represent the volatility distribution of aerosols at emission time. Current emission inventories assume POA are nonvolatile, thus in order to include a volatility distribution we use these factors. In addition, the current emission inventories do not include the intermediate volatility organic compounds (IVOCs), the sum of which makes up the additional factor of 1.5 of the original POA, based on Shrivastava et al. (2008). This was described in detail in paragraph 30 of Shrivastava et al. 2008, and it is already written in the text: "Adding up the 9 factors from each bin listed in Table 1, we obtain a total factor of 2.5, which means the new scheme's organics emission is 2.5 times that of the organics emissions in the original scheme. The additional multiplication factor of 1.5 is applied to the emission to account for missing sources of volatile organics in the IVOC volatility regime in the inventories (Shrivastava et al. 2008)."

4. **While the VBS scheme is outlined in the Model Description, it would be useful to include a brief description of how this influences, and what controls, the gas-particle partitioning among the different populations.**
Added at the end of model development section: "The amount of gas-phase species partitioned onto each aerosol population is based on the surface area of that population, in addition to the mass of that population and the volatility of species, and equilibrium partitioning is assumed."

5. **Reaction with OH is mentioned as a mechanism for chemical aging of the organic species, is the OH concentration taken from GISS ModelE? How often are the concentrations updated? How does this process occur (does it occur at a specified reaction rate for each of the volatility bins of organic species?)**
No, the OH radical concentration is calculated in the box model by using the same chemical mechanism and time step (30 minutes) as in the global model. The aging is a reaction which is taken into account in the gas-phase chemical mechanism like any other chemical reaction in the model. We added the following at the beginning of section 2: "A time step of 30 minutes is used, for consistency with the global model."

6. **Where do the products of the oxidation of biogenic organics go? Into the lowest volatility bin? Or are they treated entirely separately?**
They are treated separately, as already mentioned in page 3 21-25: "The semi-volatile nature of biogenic SOA is not represented in the VBS framework in this work. Instead, biogenic SOA are treated as non-volatile, as in the original MATRIX version, and are produced with a 10% constant yield from terpenes emissions without any requirement for oxidation before the OA is formed (Lathière et al., 2005; Tsigaridis et al., 2014). The inclusion of semi-volatile biogenic SOA will be parameterized in the same way as in the VBS framework presented here in the future."

7. **Some of the information in Section 4 seems like it would be more appropriate in the model (i.e., description of the treatment of BC) / VBS description (i.e., description of the emission rates for the VBS species), rather than the discussion of the simulations?**
We moved the description of BC in the model description, in section 2.1, and the VBS emissions factors in section 2.2. In addition, we added a short sentence here to account for the BC conversion from BC1 to BC2 that was moved elsewhere: "Condensation of VBS species on BC1 can increase the non-absorbing shell of that population, leading to formation of BC2, as described above."

8. **Page 6, lines 19 – 20: this currently sounds like each emission factor is 2.5 (which is not the case?), perhaps rephrase this to clarify what you mean.**
We agree that the presence of 2.5 here was confusing. We rephrased the sentence: "… derived from the Shrivastava et al. (2008) mass-based emission factors."

9. **In a couple of places, compounds that are low-, intermediate-, and high-volatility are referred to (e.g., Page 7, lines 13-15); in terms of the volatility bins used here, how are these categories defined?**

Added our definition at the description for VBS: "We classify organics as Murphy et al., 2014 does: low-volatility organics are in bins $10^{-2}$ to $10^{-1}$ µg m$^{-3}$ (M2 and M1 in Table 1), semi-volatile organics are in bins $10^0$ to $10^2$ µg m$^{-3}$ (M0, P1, P2), and intermediate-volatility organics are in bins $10^3$ to $10^6$ µg m$^{-3}$ (P3, P4, P5, and P6)."

10. **Page 22, Figure 2: Refer to Table 1 or Figure 1 to explain the legend entries and define OCAR**
    Fixed Figure 2 legend.

11. **Page 24/25, Figure 4/5: The layout of the columns doesn't quite match the description in the caption.**
    Corrected. Legends now read:
    "Figure 4. Temporal evolution of organic aerosol mass concentration in each organics-containing population from the new scheme (first column for January, third column for July), and the old scheme (second column for January, fourth column for July). "
    "Figure 5. Temporal evolution of organic aerosol mass concentration fraction in each organics-containing population from the new scheme (first column for January, third column for July), and the old scheme (second column for January, fourth column for July)."

12. **General comments: On several occasions, quantities are referred to as "about" ##, it would be better to either refer to a more specific value or refer to this same value as "approximate".**
    Replaced all "about" to "approximately."

13. **Specific suggestions:**
    **Page 1, line 21: should this be "high volatility" rather than "high volatile"?**
    Corrected.

    **Page 2, line 21: insert "the" between "with" and "hydroxyl"**
    Corrected.

    **Page 2, line 27: remove "it" from between "space" and "also"**
    Removed.

    **Page 2, line 28/29: specify that the "it" being referred to here is the 2D-VBS?**
    Inserted.

    **Page 3, line 23: replace "terpenes emissions" with "emissions of monoterpenes" [if you are only referring to monoterpenes here]**
    The model version we used contains both monoterpenes and higher terpenes. No changes made.

    **Page 4, line 21: remove the word "ones" from between "volatile" and "and"**
    Removed.

    **Page 5, line 11: insert a hyphen into "nonvolatile"**
    Inserted.

**Page 5, line 12-14: rephrase this sentence, it is currently not clear what you mean – and be consistent with writing numerical values as words or numbers, i.e. "eight" v. "8"**
Rephrased, now the sentence reads: "Now each of the 8 organic-containing populations carry 9 additional semi-volatile VBS species listed in Table 1. Together with the 5 original tracers, we now have up to 14 available tracers per population, depending on whether they carry organic aerosols or not, with the original organics tracer (OCAR) representing the non-volatile biogenic OA, as it did in the original mechanism."

**Page 5, line 18: you could replace "of different" with "and"?**
Replaced.

**Page 5, line 19: you could move "(Mexico City) to the end of this description (after "tropics") to avoid confusion**
Moved.

**Page 5, line 23: you could separate this into two sentences, with the first ending after "Table 3" and the second beginning with "Here we do not include. . ..."**
Separated.

**Page 6, line 30: replace "least one" with "least volatile"**
Replaced.

**Page 7, line 11: should this say "more oxidized" rather than "less oxidized"?**
Corrected.

**Page 7, line 18: replace "same with that" as "same as that"**
Replaced.

**Page 8, line 19: replace "much more" with "higher concentrations"**
Replaced.

**Page 8, line 20: replace "is also very different" with "are also very different"**
Replaced.

**Page 9, line 18: replace "effects" with "affects" Page 9, line 29: replace "reason of this" with "reason for this"**
Replaced.

1.  **Page 3 lines 21-25: This paragraph needs to be moved to the model description**
    Moved to the end of the model development section.

2.  **Section 2.1: I suggest to describe more in detail here the aerosol populations used by the MATRIX model.**
    We are now referring the description to Table 2: "MATRIX is designed to resolve the aerosol temporal evolution and represent the mixing states of a user-selected set of aerosol populations, which are modes of different composition as listed in Table 2, tracking two moments each, number and mass, while keeping the width of the distribution fixed."

3.  **Section 2.2: I suggest to improve the description of the VBS framework used.**
3.1 **Important information that affect the presented results are missing here: For instance, the description of the ageing mechanism (i.e., the oxidation rate constants used, the reduction in volatility after each oxidation step, and the oxygen mass added per oxidation).**
    Please see response to Referee #1 3.4: Gas phase organics are oxidized by OH radicals with a rate constant of $10^{-11}$ cm$^3$ s$^{-1}$, which is stated in section 3.3.1 of Donahue et al., 2006. The reduction in volatility after each oxidation step is already stated in the VBS framework description in section 2.2 of the paper, which is one bin per oxidation step.
    We do not explicitly track oxygen because that's for 2D VBS, we use OM:OC=1.6 which can be modified as needed. For simplicity, we chose to use a constant value for all volatility bins, although when we will do global modeling studies this value will be different for each bin. Choosing to use a higher OM:OC as volatility decreases would have no significant effect on results here other than a higher organic mass with decreasing volatility. Following referee 1's suggestion, we added enthalpy of vaporization in Table 1.

3.2 **Furthermore, how do you describe the formation of SOA from traditional VOCs in the model? Do you use a narrower volatility distribution (e.g., up to $10^3$)?**
    As already mentioned in the text: "The semi-volatile nature of biogenic SOA is not represented in the VBS framework in this work. Instead, biogenic SOA are treated as non-volatile, as in the original MATRIX version, and are produced with a 10% constant yield from terpenes emissions without any requirement for oxidation before the OA is formed (Lathière et al., 2005; Tsigaridis et al., 2014). The inclusion of semi-volatile biogenic SOA will be 25 parameterized in the same way as in the VBS framework presented here in the future."

3.3 **Do you use aerosol yields for the oxidation of VOCs (if so please report them)?**
    Other than the 10% from biogenic VOCs that form non-volatile SOA, which is already mentioned in the manuscript (see previous comment as well), no. All VBS-related aerosols are coming from particulate emissions that evaporated, not from gaseous precursors that got oxidized.

3.4 **Another issue is how do you perform the partition between the two phases. Do you assume instant equilibrium?**
    Yes. See also comment 3.5 from reviewer #1.

3.5 **Do you account for the temperature dependence of saturation concentrations? If so, what are the enthalpies of vaporization used for each of your organic species?**
Yes. The enthalpy of vaporization is calculated using equation 12 from Epstein et al., 2010. This is now included in Table 1. See also comment 2.1 from reviewer #1.

3.6 **Finally, since you use terms such as "high volatility range", "intermediate volatility range", and "low volatility range" in your results I would recommend to define these terms here in respect to the effective saturation concentration.**
We modified and added our definition in the VBS framework description. See also comment 5 from reviewer #1.

4. **Page 5, lines 1-4: Which is the extra population compared to the "14 populations" configuration? Furthermore, according to Table 2, the aerosol populations ACC, BC1, BC2, and BCS do not contain organics. Please explain what do you mean that these populations "are set to contain organics as semi-volatile VBS species".**
Thanks for spotting a typo in the manuscript. The original mechanism had 16 populations, not 14. This is now fixed in the revised manuscript. The one population that was eliminated is DBC (dust/BC mixtures), which we know from past work is almost always negligible. We also modified the sentence quoted by the reviewer to "could contain organics as semi-volatile VBS species". The new mechanism allows for organics to be present to more populations than the original mechanism does, namely the populations mentioned by the reviewer.

5. **Page 6 lines 1-2: Why the overestimation of biomass burning emission factors is not an issue for your experiments? Your experiments include forested areas during summer (where you have high emissions from open biomass burning) and highly populated areas during winter (where you have high biomass burning emissions from residential heating).**
This statement was not accurate. Hodzic et al. (2015) did not show that Shrivastava et al. (2008) overestimated biomass burning, instead they showed that Shrivastava et al. (2015) had the overestimation. The line has been removed.

6. **Page 7 lines 1-12: Most of this discussion belongs to the model description.**
Lines abridged and combined in the VBS part of the model description.

7. **Page 11 line 30: You did not only include the volatility of the organics, but also their reactivity (by ageing with OH).**
Yes. Now the sentence reads: "… how the inclusion of semi-volatility of organics and their reactivity affected …"

8. **Table 2: What is the size of the mode (Aitken, accumulation, or coarse) used for the aerosol populations OCC, BC1, BC2, BCS, BOC, OCS, and MXX? Also, does the OCC has sulfate? Because in that case it seems to be identical to the OCS.**
Populations can grow or shrink to the whole size distribution range, contrary to other modal models. See also reply 2 of reviewer #2. OCC could be the same in terms of composition, but has different origin: OCC is directly emitted and can be sulfate-free, while OCS comes from

coagulation of OCC with ACC and a) is a secondary particle, rather than a primary, and b) will always contain sulfate.

9.  **Figure 2: "P2" and "OCAR" seem to have very similar color. Please change the color of "P2". Furthermore, it would be nice to add for comparison the "OCAR" dashed line in the "Aerosol phase" column as well.**
    Following the recommendation of reviewer #2, we moved OCAR to the aerosol phase column. This also makes the line much more visible, and its black color does not get confused with the grey P2.

10. **Figure 4: The label is wrong. It writes that the second column is "July new" and the third for "January old" which is not the case.**
    Legend is fixed. See also comment 11 from reviewer #2.

**References:**

Bauer, S. E., Wright, D., Koch, D., Lewis, E. R., McGraw, R., Chang, L.-S., Schwartz, S. E. and Ruedy, R.: MATRIX (Multiconfiguration Aerosol TRacker of mIXing state): an aerosol microphysical module for global atmospheric models, Atmos. Chem. Phys. Discuss., 8(3), 9931–10003, doi:10.5194/acpd-8-9931-2008, 2008.

Donahue, N. M., Robinson, a. L., Stanier, C. O. and Pandis, S. N.: Coupled partitioning, dilution, and chemical aging of semivolatile organics, Environ. Sci. Technol., 40(8), 2635–2643, doi:10.1021/es052297c, 2006.

Epstein, S. A., Riipinen, I., and Donahue, N. M.: A Semi-Empirical Correlation between Enthalpy of Vaporization and Saturation Concentration for Organic Aerosol, Environ. Sci. Technol., 44, 743–748, doi:10.1021/es902497z, 2010.

Hodzic, A., Kasibhatla, P. S., Jo, D. S., Cappa, C., Jimenez, J. L., Madronich, S., and Park, R. J.: Rethinking the global secondary organic aerosol (SOA) budget: stronger production, faster removal, shorter lifetime, Atmos. Chem. Phys. Discuss., 15, 32413-32468, doi:10.5194/acpd-15-32413-2015, 2015.

Murphy, B. N., Donahue, N. M., Robinson, A. L., and Pandis, S. N.: A naming convention for atmospheric organic aerosol, Atmos. Chem. Phys., 14, 5825-5839, doi:10.5194/acp-14-5825-2014, 2014.

Pankow, J. F.: An absorption-model of gas-particle partitioning of organic compounds in the atmosphere, Atmos. Environ., 28, 185–188, 1994.

Shrivastava, M. K., Lane, T. E., Donahue, N. M., Pandis, S. N. and Robinson, A. L.: Effects of gas particle partitioning and aging of primary emissions on urban and regional organic aerosol concentrations, J. Geophys. Res. Atmos., 113(18), doi:10.1029/2007JD009735, 2008.

Shrivastava, M., Easter, R., Liu, X., Zelenyuk, A., Singh, B., Zhang, K., Ma, P-L, Chand, D., Ghan, S., Jimenez, J. L., Zhang, Q., Fast, J., Rasch, P., and Tiitta, P.: Global transformation and fate of SOA: Implications of low volatility SOA and gasphase fragmentation reactions, J. Geophys. Res.-Atmos., 120, 4169–4195, doi:10.1002/2014JD022563, 2015.

---

## Author Response (AR2)

Dear Editor,

Thank you for accepting the paper for publication to GMD. We followed your recommendation and reverted the title back to the one used in GMDD. In addition, we now use full first names of all authors, as requested in the GMD upload page.

Sincerely,

C. Y. Gao, K. Tsigaridis and S. E. Bauer

[revised manuscript text omitted]